# Jak2-mediated phosphorylation of Atoh1 is critical for medulloblastoma growth

Tiemo J Klisch[1,2]*, Anna Vainshtein[1,2], Akash J Patel[1,3], Huda Y Zoghbi[1,2,3,4]

[1]Jan and Dan Duncan Neurological Research Institute, Texas Children's Hospital, Houston, United States; [2]Department of Molecular and Human Genetics, Baylor College of Medicine, Houston, United States; [3]Department of Neurosurgery, Baylor College of Medicine, Houston, United States; [4]Howard Hughes Medical Institute, Baylor College of Medicine, Houston, United States

**Abstract** Treatment for medulloblastoma, the most common malignant brain tumor in children, remains limited to surgical resection, radiation, and traditional chemotherapy; with long-term survival as low as 50–60% for Sonic Hedgehog (Shh)-type medulloblastoma. We have shown that the transcription factor Atonal homologue 1 (Atoh1) is required for Shh-type medulloblastoma development in mice. To determine whether reducing either Atoh1 levels or activity in tumors after their development is beneficial, we studied Atoh1 dosage and modifications in Shh-type medulloblastoma. Heterozygosity of Atoh1 reduced tumor occurrence and prolonged survival. We discovered tyrosine 78 of Atoh1 is phosphorylated by a Jak2-mediated pathway only in tumor-initiating cells and in human SHH-type medulloblastoma. Phosphorylation of tyrosine 78 stabilizes Atoh1, increases Atoh1's transcriptional activity, and is independent of canonical Jak2 signaling. Importantly, inhibition of Jak2 impairs tyrosine 78 phosphorylation and tumor growth in vivo. Taken together, inhibiting Jak2-mediated tyrosine 78 phosphorylation could provide a viable therapy for medulloblastoma.
DOI: https://doi.org/10.7554/eLife.31181.001

## Introduction

Medulloblastoma (MB) is the most common malignant primary brain tumor in children (*Dhall, 2009*) and is thought to originate from rhombic lip-derived precursors that have failed to differentiate into their usual progeny (*Hatten and Roussel, 2011*). Molecular studies have allowed us to classify MB into four major subtypes: WNT, sonic hedgehog (SHH), Group C, and Group D (*Taylor et al., 2012*). The cell of origin differs among the subgroups, as does the clinical prognosis (*Taylor et al., 2012*). SHH-type tumors arise from precursors of the cerebellar granule neurons and can contain mutations in Smoothened, Patched, or other genes in this pathway (*Rimkus et al., 2016*). Elevated expression of Atoh1, the basic helix-loop-helix transcription factor that is crucial for the development of the cerebellar granule neurons (*Ben-Arie et al., 1996*), can also drive uncontrolled cell proliferation and formation of medulloblastoma (*Flora et al., 2009*). In addition to attempting to inhibit SHH signaling—a strategy that can be foiled by the development of secondary mutations (*Yauch et al., 2009*)—we hypothesized that controlling the levels of Atoh1 protein will provide another therapeutic entry point for the treatment SHH-driven medulloblastoma.

## Results and discussion

To determine the influence of Atoh1 gene dosage on MB development and prognosis, we examined the rate of tumor formation in a mouse model of Shh-dependent MB, the Nd2::SmoA1[Tg] line (*Hatton et al., 2008*). These mice have a constitutively active Shh pathway in all cerebellar granule

*For correspondence:
klisch@bcm.edu

**eLife digest** Medulloblastoma is the most common solid brain tumor that develops in children, with more than five hundred new cases diagnosed in the United States every year. There are four broad types of medulloblastoma. One of these is called the "Sonic Hedgehog" subtype, named after the biological pathway that becomes re-activated in these tumors. Only about half of patients with this subtype survive for more than 10 years. Moreover, medulloblastoma treatment combines surgery, chemotherapy and radiation, which can cause severe side effects including psychiatric disorders and cognitive impairment.

Several drugs that treat medulloblastoma by targeting the Sonic Hedgehog pathway are currently being tested in clinical trials. However, these drugs are usually only effective for a limited time before the tumor evades the treatment. Therefore, there is a need to develop new treatment options for medulloblastoma, perhaps by targeting different signaling pathways in the cells.

A protein called Atoh1 is needed for proper brain development in humans, but is not normally present after the first year of life. This protein is, however, re-expressed at high levels in medulloblastoma in mice and humans and is essential for Sonic Hedgehog-type medulloblastoma to form in mice.

Klisch et al. used genetic techniques to reduce the amount of Atoh1 in mice that develop medulloblastoma. This intervention reduced the number of mice that developed tumors and increased their lifespan. Biochemical experiments showed that the tumor stem cells of the mice contain a modified version of Atoh1 where a phosphate molecule is bound to a particular region of the protein. This phosphorylation increased the amount and activity of Atoh1 in the cell, and so caused tumors to grow more quickly in mice. Phosphorylated Atoh1 was also detected in samples taken from human medulloblastoma tumors.

Klisch et al. also found that an enzyme called Jak2 phosphorylates Atoh1. Inhibiting Jak2 reduced the levels of Atoh1 in medulloblastoma cells and slowed tumor growth in mice. Future work could investigate different ways of preventing Atoh1 phosphorylation, with the hope of finding new treatments for Sonic-Hedgehog-type medulloblastomas.

DOI: https://doi.org/10.7554/eLife.31181.002

cells and spontaneously develop MBs (*Hatton et al., 2008*). Whereas almost a quarter (23.1%) of Nd2::SmoA1$^{Tg}$ mice developed MB and succumbed to the disease within 300 days of life, fewer than one in ten (6.7%) Nd2::SmoA1$^{Tg}$ mice – heterozygous for *Atoh1* (Atoh1$^{+/-}$, Nd2::SmoA1$^{Tg}$) – developed signs of disease (*Figure 1a*, *Figure 1—source data 1* for all statistical analysis). *Atoh1* heterozygosity dropped the overall tumor incidence by over three-fold and never reached that of Nd2::SmoA1$^{Tg}$, *Atoh1* wildtype mice. Importantly, while both groups expressed the tumorigenic transgene (Nd2::SmoA1) at similar levels (*Figure 1—figure supplement 1a*), levels of Gli2, the primary Shh mediator in medulloblastoma and direct target of Atoh1 (*Read et al., 2009*; *Klisch et al., 2011*) were reduced in the Atoh1 heterozygous mice. Hence, a 50% reduction in Atoh1 protein level has a direct, positive, effect on survival in this medulloblastoma mouse model. This suggests that Atoh1 protein dosage is critical for the development of medulloblastoma in mice.

The regulation of Atoh1 expression during the development of the cerebellum is highly complex, involving not only Shh signaling but also other signaling pathways required for proper development, such as BMP and WNT signaling (*Butts et al., 2014*). While this regulation is transcriptional in nature, another layer of regulation has been reported to influence Atoh1 protein levels: phosphorylation (*Forget et al., 2014*). Hence, we asked whether Atoh1 was aberrantly phosphorylated in the tumors. Using tumors isolated from Nd2::SmoA1$^{Tg}$ mice, we performed IP-MS on tumor-initiating cells that express Atoh1 [marked by cell surface protein CD15 (*Read et al., 2009*)] and CD15-negative cells (*Figure 1—figure supplement 1b*). We tested the integrity of both cell populations by flank injections of sorted cells in immunosuppressed mice and found that *Atoh1* expression in the tumor-initiating cells is required to drive secondary tumor formation (*Figure 1—figure supplement 1b*). Using Atoh1 IP-MS, we discovered that tyrosine 78 (Y78, *Figure 1—source data 2*) on Atoh1 is phosphorylated exclusively in tumor-initiating cells. We confirmed phosphorylation of this Atoh1 residue in MB tissue using a phospho-specific antibody (p-Atoh1-Y78, *Figure 1b*). We further confirmed

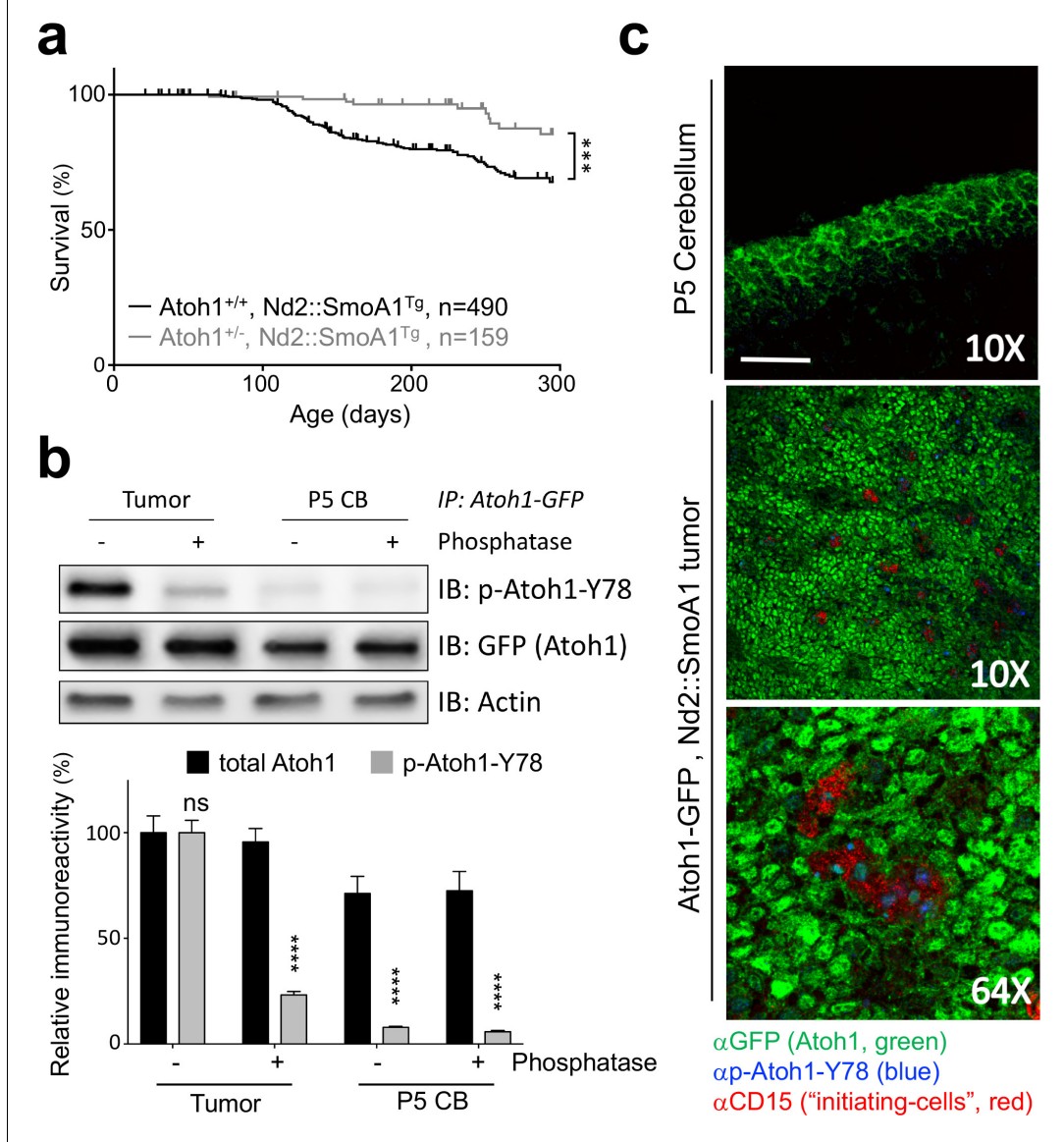

**Figure 1.** Tyrosine 78 of Atoh1 protein is phosphorylated in MB-initiating cells in vivo. (a) Kaplan-Meier survival curve shows that Atoh1 heterozygosity markedly reduces MB tumor incidence in mice (Log Rank (Mantel-Cox)). (b) Atoh1 tyrosine 78 is strongly phosphorylated in malignant but not healthy cerebellar tissue (n = 3, representative blot shown, Two-way ANOVA with Sidak's multiple comparisons test). (c) Y78 (blue) is phosphorylated in the tumor-initiating cell (CD15+, red) islets. Values are mean ± s.e.m.; ***p<0.001, ****p<0.0001, scale = 200 μm.
DOI: https://doi.org/10.7554/eLife.31181.003

The following source data and figure supplement are available for figure 1:

**Source data 1.** This excel file contains all relevant statistical analyses for the manuscript.
DOI: https://doi.org/10.7554/eLife.31181.005
**Source data 2.** Atoh1 tyrosine 78 coverage and Atoh1 interacting proteins.
DOI: https://doi.org/10.7554/eLife.31181.006
**Figure supplement 1.** Atoh1 is required for tumor-initiating cells.
DOI: https://doi.org/10.7554/eLife.31181.004

these findings in tumorigenic and control tissues from mice and found that Atoh1 Y78 phosphorylation is specific to the tumor-initiating cells (*Figure 1c*). Y78 immuno-reactivity was observed mostly in the initiating cell islets of the tumor; neither the Y78-specific phospho-antibody, nor an antibody

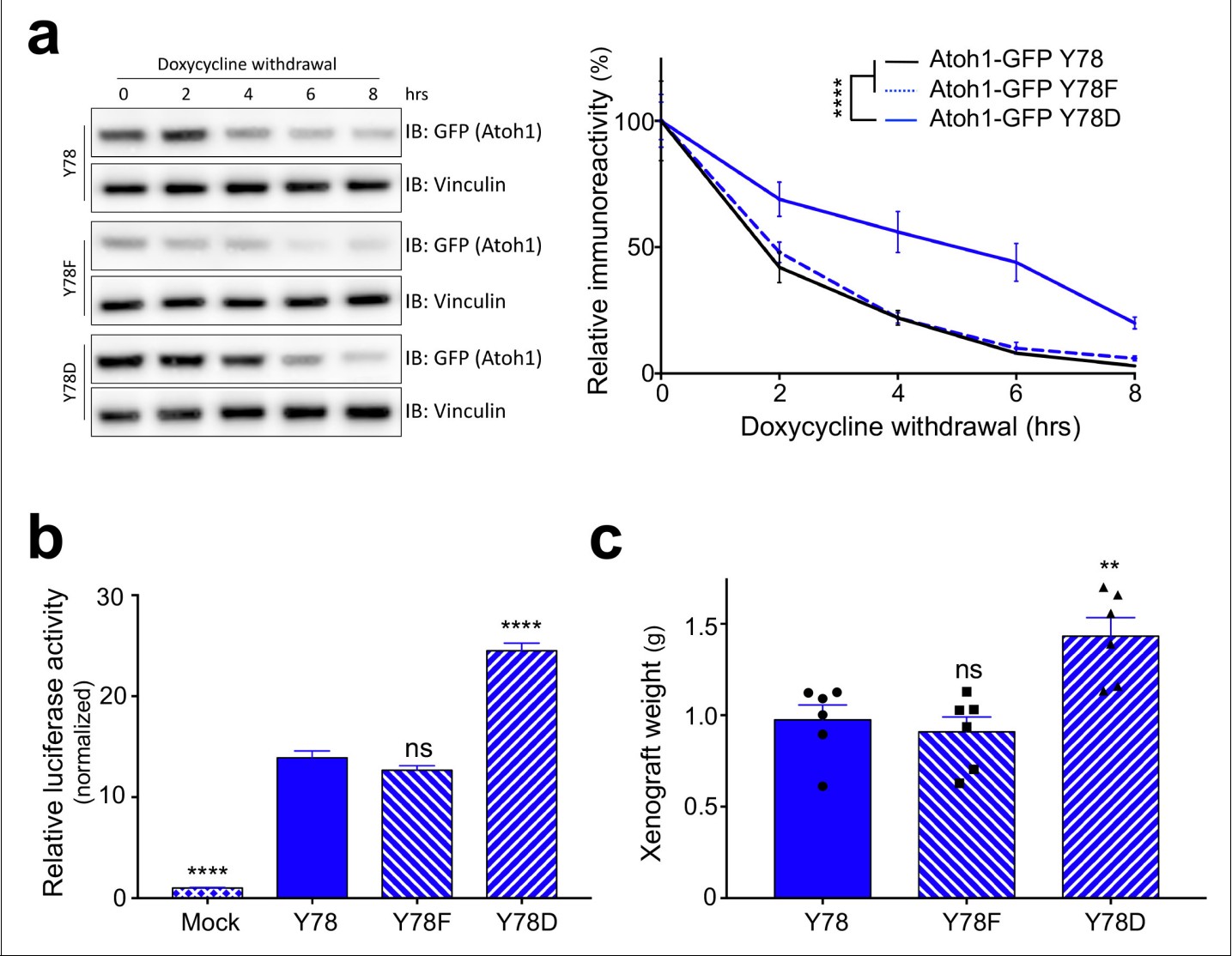

**Figure 2.** Y78 phosphorylation regulates Atoh1 levels. (**a**) Y78 phosphorylation increases Atoh1 stability in doxycycline inducible, stable DAOY cell lines (n = 3, representative blot shown, non-linear regression). (**b**) Y78 phosphorylation increases transcriptional activity of Atoh1-specific reporter in DAOY cells (n = 46 in duplicates, One-way ANOVA compared to Y78, Dunnett's multiple comparisons test). (**c**) Y78 phosphorylation significantly increases tumor growth in grafted, Atoh1-transduced DAOY cells three months after implantation (n = 6, One-way ANOVA compared to Y78). Y78 = Atoh1-GFP WT, Y78F = Atoh1-GFP Y78F, Y78D = Atoh1-GFP Y78D; values are mean ± s.e.m.; **p<0.01, ****p<0.0001, ns = not significant.
DOI: https://doi.org/10.7554/eLife.31181.007

The following source data and figure supplements are available for figure 2:

**Source data 1.** Atoh1 half-life analysis based on stably expressing cell lines.
DOI: https://doi.org/10.7554/eLife.31181.010
**Figure supplement 1.** Tyrosine 78 molecular consequences.
DOI: https://doi.org/10.7554/eLife.31181.008
**Figure supplement 2.** Tyrosine 78 biological consequences.
DOI: https://doi.org/10.7554/eLife.31181.009

against CD15, reacted in the normal tissue, even in the presence of strong Atoh1 expression in the external granule layer (*Figure 1c*).

Because Atoh1 is not endogenously expressed in neuronal cell lines, and reliable primary cultures of human SHH-type medulloblastoma cells are not established, we turned to modify and study medulloblastoma cell lines that have been used in the past. We engineered cell lines expressing

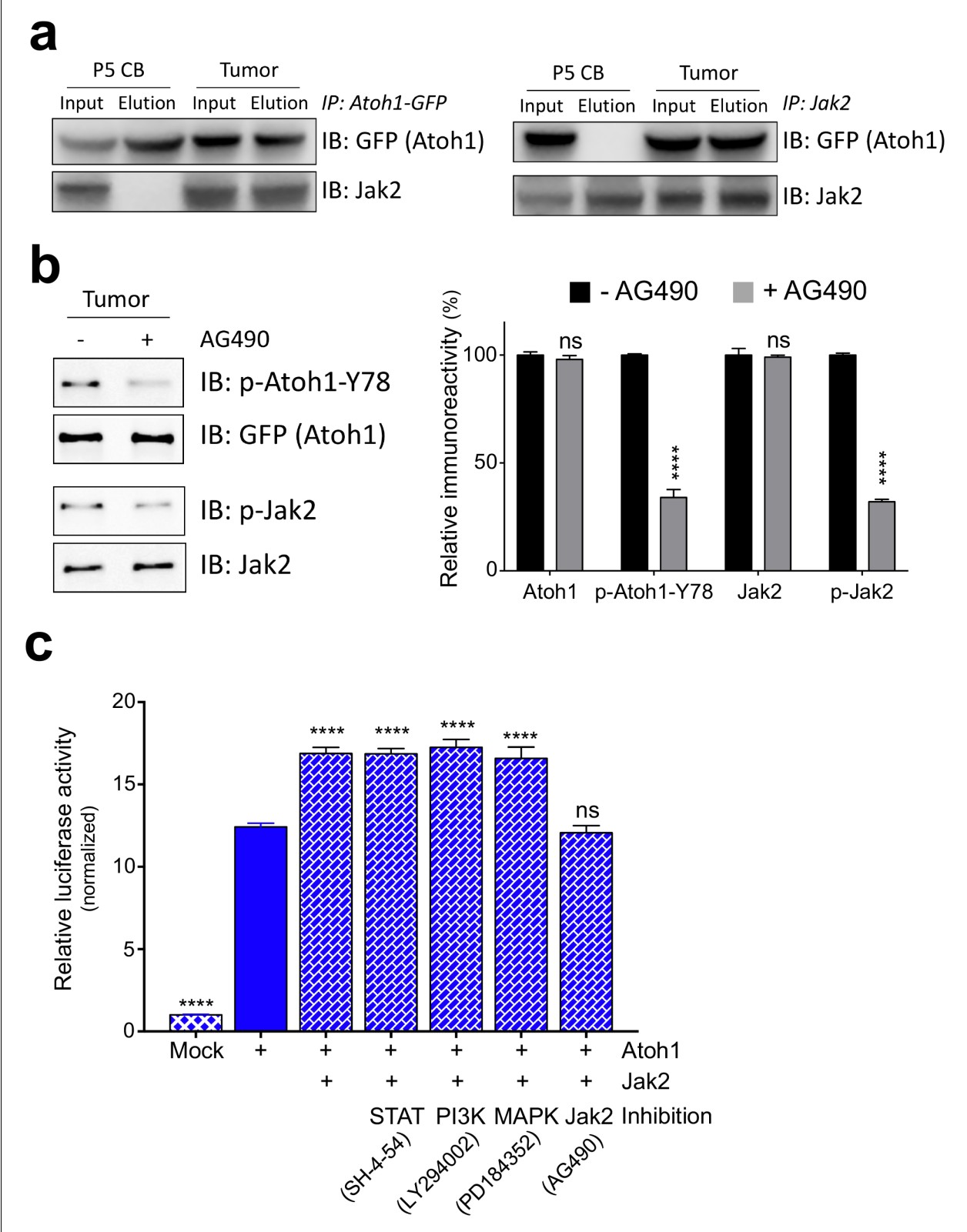

**Figure 3.** Jak2 phosphorylates Atoh1 Y78. (a) IP of Jak2 or Atoh1 in tumor and P5 cerebellar tissue shows that Atoh1 and Jak2 interact in tumor tissue. (b) Upon inhibition of Jak2, Y78 phosphorylation levels fall dramatically in ex vivo culture (n = 3, Two-way ANOVA with Sidak's multiple comparisons test). (c) Transcriptional activity of Atoh1 is increased upon Jak2 introduction, independent of canonical Jak2 pathways (n = 46 in duplicates, One-way ANOVA with Dunnett's multiple comparisons test compared to Atoh1-GFP). Values are mean ± s.e.m.; ****p<0.0001, ns = not significant.

*Figure 3 continued on next page*

*Figure 3 continued*

DOI: https://doi.org/10.7554/eLife.31181.011

The following figure supplements are available for figure 3:

**Figure supplement 1.** Jak2 phosphorylates tyrosine 78.

DOI: https://doi.org/10.7554/eLife.31181.012

**Figure supplement 2.** Jak2 increases Atoh1 stability.

DOI: https://doi.org/10.7554/eLife.31181.013

**Figure supplement 3.** Jak2 increases proliferation via tyrosine 78.

DOI: https://doi.org/10.7554/eLife.31181.014

either doxycycline-inducible wildtype (Y78), phospho-dead (Y78F) or phospho-mimetic (Y78D) Atoh1, using three different human-derived MB parental cell lines: DAOY (marked in blue), UW228 (orange) and ONS-76 (purple) (*Zanini et al., 2013*). While these cell lines have been used extensively, it has been recently suggested that at least the DAOY cell line is more 'Shh-like' than the others (*Higdon et al., 2017*). We found that Y78 phosphorylation status did not alter the interaction of Atoh1 with its E-protein binding partner or its nuclear-cytoplasmic localization (*Figure 2—figure supplement 1a–b*). However, upon doxycycline withdrawal, Y78 phosphorylation dramatically increased Atoh1 protein stability in all three parental lines (*Figure 2a*, *Figure 2—figure supplement 1c–d*). Protein half-life of phospho-mimetic Atoh1 was triple that of wildtype Atoh1 (*Figure 2—source data 1*), while the RNA stability of the transgenes was comparable (*Figure 2—figure supplement 1e*).

To test the transcriptional activity of the Atoh1 variants we used an Atoh1-specific luciferase reporter system (*Klisch et al., 2011*). Overexpression of phospho-mimetic Atoh1 (Y78D), but not the phospho-mutant (Y78F), increased Atoh1 transcriptional activity (*Figure 2b*, *Figure 2—figure supplement 2b–c*). In a proliferation assay using all three parental cell lines, we found that the enhanced activity of Atoh1 Y78D resulted in hyper-proliferation, whereas the phospho-dead Atoh1 resulted in a slight decrease in cell proliferation (*Figure 2—figure supplement 2d–f*). While the increase in transcriptional activity could be due to the increased half-life, the steady-state levels of the three Atoh1 mutants were similar in the luciferase lysates (*Figure 2—figure supplement 2a*). These data argue for two distinct effects of tyrosine 78 phosphorylation: increased Atoh1 stability as well as increased Atoh1 transcriptional activity.

To test whether this phosphorylation drives tumor growth in vivo, we injected DAOY cells [known to form secondary tumors (*Jacobsen et al., 1985*)] stably expressing doxycycline-inducible Atoh1-GFP WT, Atoh1-GFP Y78F, or Atoh1-GFP Y78D in flank xenografts. While these flank xenograft models lack the fully functional immune system, they provide a useful model for longitudinal tumor growth monitoring with minimal burden to the animal. The mice were fed doxycycline chow continuously to express the transgenes. Expression of wildtype or phospho-dead Atoh1 (Y78F) led to a similar average tumor weight, but overexpression of phospho-mimetic Atoh1 (Y78D) increased tumor weight by roughly 40% (*Figure 2c*). These data suggest that tyrosine 78 phosphorylation increases Atoh1 transcriptional activity, which in turn increases cellular proliferation in vitro as well as in vivo.

To identify the tyrosine kinase responsible for the aberrant phosphorylation of Atoh1 in the malignant cells, we revisited our IP-MS data. We recovered 637 Atoh1-interacting proteins from CD15-positive cells and 278 interactors from CD15-negative cells (*Figure 1—source data 2*). As would be expected, E-proteins were the strongest interactors present in all pull down assays, but interestingly only one tyrosine kinase, Janus kinase 2 (Jak2), was present exclusively in tumor-initiating cells. The Jak2 paralog Jak1 was present in the non-initiating cell population (*Figure 1—source data 2*). The Janus kinases typically function in the cytoplasm, downstream of cytokine signaling, and activate canonical cascades such as signal transducer and activator of transcription (STAT), mitogen activated protein kinase (MAPK), and/or the phosphotidylinositol 3-kinase (PI3K)–Akt pathways (*Levine et al., 2007*). Although one of the most common gain-of-function mutations in Jak2 (Jak2$^{V617F}$) is associated with increased proliferation in human myeloproliferative diseases (*Levine et al., 2007*), Jak2 function in medulloblastoma has not been investigated.

We tested Jak1-3 and Tyrosine Kinase 2 (Tyk2), as well as Protein Tyrosine Kinase 2 (Ptk2) in a bimolecular fluorescence complementation assay (BiFC) for an Atoh1 interaction (*Neet and Hunter,*

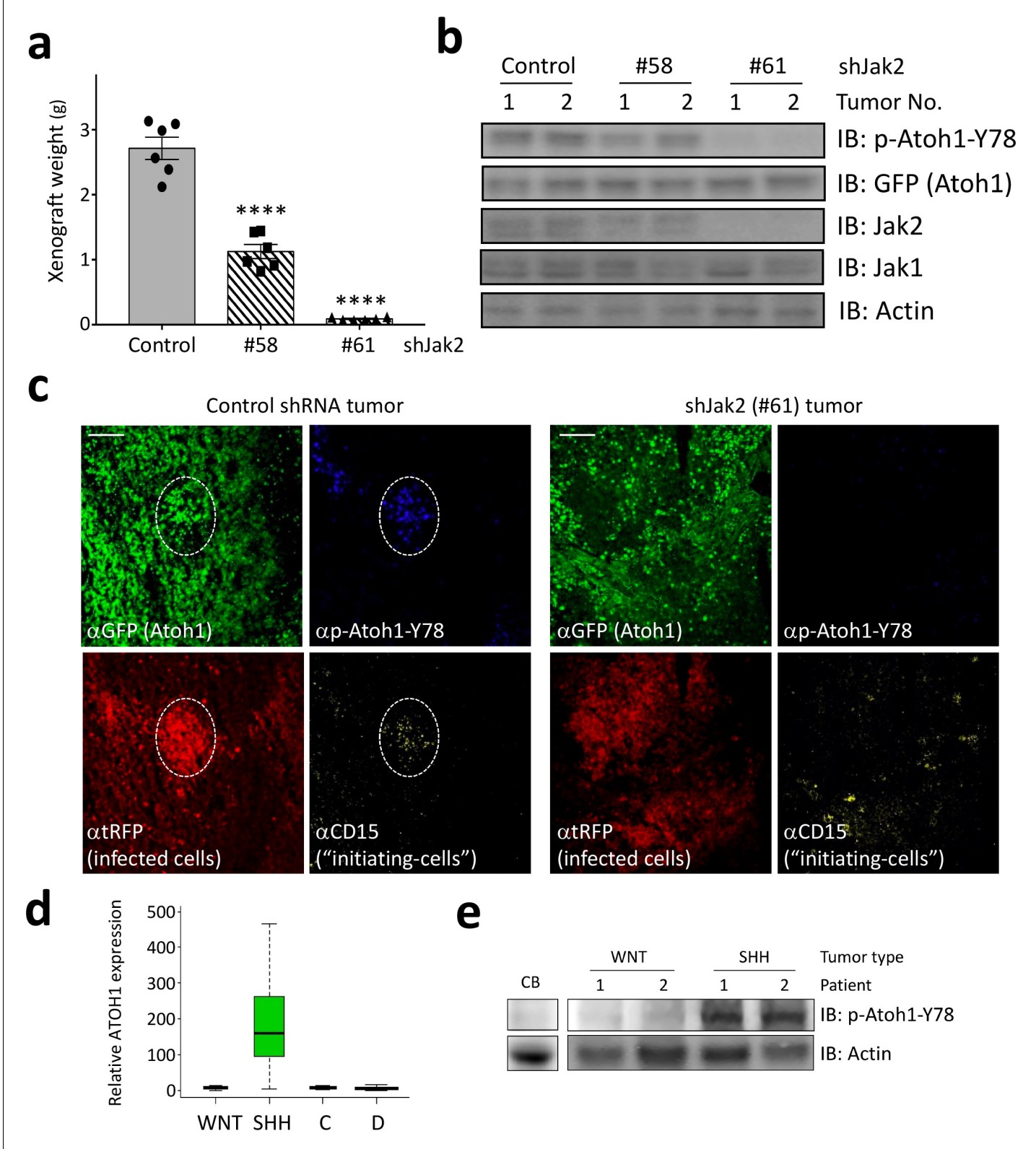

**Figure 4.** Jak2 inhibition reduces tumor growth in vivo. (a–b) In xerograph experiments, inhibition of Jak2 via shRNA inhibits tumor growth (n = 6, One-way ANOVA with Dunnett's multiple comparisons test) (a), which is paralleled by loss of Y78 phosphorylation (b). (c) shJak2-infected tumor cells display neither defined initiating islets nor Y78 phosphorylation. (d) Box plot of human ATOH1 expression showing that Atoh1 is expressed only in the SHH

*Figure 4 continued on next page*

Figure 4 continued

subtype and no other subtypes. Data were taken from *Kool et al., 2012*. (e) Human MB tissue displays Y78 phosphorylation in SHH but not WNT-type tumors or post-mortem adult cerebellar tissue. Values are mean ± s.e.m.; ****p<0.0001, scale = 200 µm.

DOI: https://doi.org/10.7554/eLife.31181.015

*1996*). Only Jak2 displayed a strong interaction with Atoh1 as indicated by the shift in GFP signal detected by flow cytometry (*Figure 3—figure supplement 1a*). We further investigated whether these kinases were able to phosphorylate Y78 using a luciferase-based kinase assay. Once again, only Jak2 was able to phosphorylate wildtype Atoh1 (Y78) in a dose dependent manner in vitro (*Figure 3—figure supplement 1b*). This phosphorylation was specific to Y78, as the addition of phospho-mutant Atoh1 (Y78F) abolished this effect (*Figure 3—figure supplement 1b*). Given that both Atoh1 and Jak2 are expressed during cerebellar development and both are highly overexpressed in MB tissue (*Figure 3—figure supplement 1c*), we tested whether the Jak2-Atoh1 interaction occurs in vivo. We immunoprecipitated Atoh1 or Jak2 and found a strong interaction in malignant but not normal cerebellar tissue (*Figure 3a*). To determine whether Jak2 is a critical kinase phosphorylating tyrosine 78, we isolated Shh-type MB cells and cultured them in the presence or absence of a strong and highly specific Jak2 inhibitor (AG490) (*Kobayashi et al., 2015*). One-hour treatment with the inhibitor was sufficient to drastically reduce Jak2 and Atoh1 Y78 phosphorylation (*Figure 3b*). These findings demonstrate that Jak2 is activated in Shh-type MB, where it interacts with Atoh1 and phosphorylates it on tyrosine 78.

We examined whether Jak2 influences Atoh1 activity through its canonical downstream cascades by using the Atoh1 transcriptional reporter assay. Co-transfection of Atoh1 and Jak2 dramatically increased Atoh1 activity in all three MB lines (*Figure 3c*, *Figure 3—figure supplement 1d–e*). Inhibiting Jak2 with AG490 abolished this, but blocking STAT, PI3K or MAPK signaling in this assay had no effect (*Figure 3c*, *Figure 3—figure supplement 1d–e*), arguing that Jak2 influences Atoh1 directly. Prior studies have argued for non-canonical roles of Jak2, e.g. Jak2 phosphorylates tyrosine 41 on histone H3 (*Dawson et al., 2009*). Moreover, most of the genes regulated by Jak2 do not contain the predicted STAT DNA-binding motifs (*Ghoreschi et al., 2009*) suggesting the existence of unidentified transcriptional mediators. Overexpression of Jak2 pheno-copied the transcriptional gain-of-function phenotype of Atoh1 Y78D, increased the stability of wildtype Atoh1 (Y78) to Y78D levels (*Figure 3—figure supplement 2*, *Figure 2—source data 1*) and increased cellular proliferation in vitro, which was blocked by phospho-mutant Atoh1 (Y78F) and the addition of AG490 (*Figure 3—figure supplement 3*).

Having identified Jak2 as a critical kinase phosphorylating tyrosine 78, we asked if inhibition of Jak2 might be a viable strategy to decrease Atoh1 levels in vivo. We isolated Shh-type medulloblastoma cells, infected them with lentiviral particles harboring Jak2 shRNAs tagged with RFP, and grafted these into mice. Control shRNA-infected tumors grew rapidly, but both Jak2 shRNA-infected tumors showed dramatically reduced tumor growth (*Figure 4a*), in accordance with their potency to inhibit Jak2 (*Figure 4b*). The decrease in Jak2 resulted in a concomitant decrease in Y78 phosphorylation (*Figure 4b*). Moreover, Jak1, a close relative of Jak2, did not compensate for the inhibition of Jak2 as it was expressed at similar levels in control and Jak2 inhibited tumors (*Figure 4b*). Sections from control shRNA-infected tumors revealed large areas of infected cells (*Figure 4c* left panel, red) that expressed Atoh1 and had pockets of tumor-initiating cells (CD15-positive, yellow) that co-localized with phosphorylated Y78 (blue). In contrast, tumors infected with the most potent Jak2 shRNA revealed scattered initiating cell clusters, none of which contained Y78 phosphorylated Atoh1 (*Figure 4c*, right panel).

Taken together, here we demonstrate that Atoh1, a protein not expressed in the post-natal normal brain, is expressed specifically in SHH-type medulloblastoma malignancies (*Figure 4d*), and that tyrosine 78 phosphorylation by Jak2 increases Atoh1 activity and stability, thus augmenting the proliferation of tumor-initiating cells in MB. Our results suggest that targeting Atoh1 by inhibiting its phosphorylation on tyrosine 78 presents a viable avenue for targeting MB cells without impinging on healthy brain tissue. Moreover, tyrosine 78 of Atoh1 can be detected in protein lysates of human samples of SHH-type medulloblastoma (*Figure 4e*), further supporting the clinical relevance of these findings.

# Materials and methods

## Key resources table

| Reagent type (species) or resource | Designation | Source or reference | Identifiers | Additional information |
|---|---|---|---|---|
| DAOY cell line (human) | inducible Atoh1-GFP WT | this paper | N/A | transgenic cell line which expresses mouse Atoh1 mutants or wildtype, tagged with GFP upon dox treatment. |
| DAOY cell line (human) | inducible Atoh1-GFP Y78F | this paper | N/A | transgenic cell line which expresses mouse Atoh1 mutants or wildtype, tagged with GFP upon dox treatment. |
| DAOY cell line (human) | inducible Atoh1-GFP Y78D | this paper | N//A | transgenic cell line which expresses mouse Atoh1 mutants or wildtype, tagged with GFP upon dox treatment. |
| UW228 cell line (human) | inducible Atoh1-GFP WT | this paper | N/A | transgenic cell line which expresses mouse Atoh1 mutants or wildtype, tagged with GFP upon dox treatment. |
| UW228 cell line (human) | inducible Atoh1-GFP Y78F | this paper | N/A | transgenic cell line which expresses mouse Atoh1 mutants or wildtype, tagged with GFP upon dox treatment. |
| UW228 cell line (human) | inducible Atoh1-GFP Y78D | this paper | N/A | transgenic cell line which expresses mouse Atoh1 mutants or wildtype, tagged with GFP upon dox treatment. |
| ONS-76 cell line (human) | inducible Atoh1-GFP WT | this paper | N/A | transgenic cell line which expresses mouse Atoh1 mutants or wildtype, tagged with GFP upon dox treatment. |
| ONS-76 cell line (human) | inducible Atoh1-GFP Y78F | this paper | N/A | transgenic cell line which expresses mouse Atoh1 mutants or wildtype, tagged with GFP upon dox treatment. |
| ONS-76 cell line (human) | inducible Atoh1-GFP Y78D | this paper | N/A | transgenic cell line which expresses mouse Atoh1 mutants or wildtype, tagged with GFP upon dox treatment. |
| recombinant DNA reagent | Atoh1-GFP_pcDNA3mod | this paper | N/A | Construct for overexpression in cell culture |
| recombinant DNA reagent | Atoh1-GFP-Y78F_pcDNA3mod | this paper | N/A | Construct for overexpression in cell culture |
| recombinant DNA reagent | Atoh1-GFP-Y78D_pcDNA3mod | this paper | N/A | Construct for overexpression in cell culture |
| recombinant DNA reagent | E47-FLAG_pcDNA3mod | this paper | N/A | Construct for overexpression in cell culture |
| recombinant DNA reagent | E2.2-FLAG_pcDNA3mod | this paper | N/A | Construct for overexpression in cell culture |
| recombinant DNA reagent | 3x AtEAM_pGL4.23 | this paper | N/A | Luciferase reporter construct |
| recombinant DNA reagent | Atoh1-GFP_pINDUCER | this paper | N/A | Construct for virus production |
| recombinant DNA reagent | Atoh1-GFP_Y78F_pINDUCER | this paper | N/A | Construct for virus production |
| recombinant DNA reagent | Atoh1-GFP-Y78D_pINDUCER | this paper | N/A | Construct for virus production |
| recombinant DNA reagent | JaK2-FLAG_pcDNA3mod | this paper | N/A | Construct for overexpression in cell culture |
| recombinant DNA reagent | Atoh1-nYFP_pBiFC | this paper | N/A | Construct for overexpression in cell culture for interaction analysis |
| recombinant DNA reagent | Jak1-cYFP_pBiFC | this paper | N/A | Construct for overexpression in cell culture for interaction analysis |
| recombinant DNA reagent | Jak2-cYFP_pBiFC | this paper | N/A | Construct for overexpression in cell culture for interaction analysis |
| recombinant DNA reagent | Jak3-cYFP_pBiFC | this paper | N/A | Construct for overexpression in cell culture for interaction analysis |
| recombinant DNA reagent | Tyk2-cYFP_pBiFC | this paper | N/A | Construct for overexpression in cell culture for interaction analysis |
| recombinant DNA reagent | Ptk2-cYFP_pBiFC | this paper | N/A | Construct for overexpression in cell culture for interaction analysis |

## Previously described protocols

All described assays as well as standard molecular biology techniques (RNA extraction, qRT-PCR, general cell culture, cell lysis, co-immunoprecipitation (co-IP), nuclear-cytoplasmic fractionation and immunoblotting) can be requested in step-by-step, 'protocol-style' format. Plasmids used in this study are available upon request (see Key Resource table). The following assays were described elsewhere: virus production and stable cell line engineering (*Rousseaux et al., 2016*), Atoh1-specific dual luciferase reporter assay (*Klisch et al., 2011*), immunofluorescence staining (*Flora et al., 2009*), BiFC (*Pusch et al., 2011*), ADP glow in vitro kinase assay (*Ohana et al., 2011*).

## Mouse models

Generation of mouse strains and genotyping procedures were previously described: *Atoh1* lacZ (*Ben-Arie et al., 2000*) (*B6.129S7-Atoh1^{tm2Hzo}/J*, RRID:IMSR_JAX:005970), Atoh1-GFP (*Rose et al., 2009*) (*B6.129S-Atoh1^{tm4.1Hzo}/J*, RRID:IMSR_JAX:013593), Nd2::SmoA1 (*Hatton et al., 2008*) (*C57BL/6-Tg(Neurod2-Smo*A1)199Jols/J*, RRID:IMSR_JAX:008831) and NOD/SCID (*Shultz et al., 1995*) (*NOD.CB17-Prkdc^{scid}/J*, RRID:IMSR_JAX:001303) recipient. All mice in this study were Nd2::SmoA1 C57/Bl6 on either Atoh1-GFP homozygous (*Nd2::SmoA1^{Tg} Atoh1^{Atoh1-GFP/Atoh1-GFP}*, 100% Atoh1 protein) or Atoh1 heterozygous (*Nd2::SmoA1^{Tg} Atoh1^{Atoh1-GFP/LacZ}*, 50% Atoh1 protein) background. All procedures were approved in advance under the guidelines of the Center for Comparative Medicine, Baylor College of Medicine and were performed in accordance with the National Institutes of Health Guide for the Care and Use of Laboratory Animals. Both male and female mice were used. Mice were group-housed in temperature-controlled rooms on a 14–10 hr light–dark cycle, at constant temperature (22–24°C) and received food and water *ad libitum*. Mice of both sexes were randomly assigned to different treatment groups.

## Human subjects

All patients provided written informed consent and tissues were collected under an IRB approved protocol at Baylor College of Medicine (BCM).

## Antibodies

*Western blot* (overnight incubation with a 1:5000 dilution): anti-p-Atoh1-Y78 (generated by GeneScript, Piscataway, NJ upon our request), anti-GFP (GeneTex, Irvine, CA Cat# GTX26556 RRID:AB_371421), anti-Jak2 (Cell Signaling Technology, Danvers, MA Cat# 3230, RRID:AB_2128522), anti-p-Jak2 (Tyr1007/1008, Cell Signaling Technology Cat# 3771S, RRID:AB_330403), anti-Vinculin (1:20,000, Sigma-Aldrich, St. Louis, MO Cat# V9131 RRID:AB_477629), anti-beta Actin-HRP (1:40,000, Abcam, Cambridge, MA Cat# ab20272 RRID:AB_445482), anti-Rabbit-HRP (1:20,000, Bio-Rad/AbD Serotec, Hercules, CA Cat# 170–5046 RRID:AB_11125757), anti-FLAG M2-HRP (1:10,000, Sigma-Aldrich Cat# A8592 RRID:AB_439702), anti-Mouse-HRP (1:50:000, Jackson ImmunoResearch Labs, West Grove, PA Cat# 715-035-150 RRID:AB_2340770); *Immunofluorescence*: anti-p-Atoh1-Y78 (1:50), anti-GFP (1:200, Abcam Cat# ab13970 RRID:AB_300798), anti-RFP (1:100, RF5R, Abcam Cat# ab125244 RRID:AB_10973556), anti-CD15-PE (1:50, BD Biosciences, San Jose, CA Cat# 347420 RRID:AB_400298), anti-chicken IgY-Alexa647 (1:100, Thermo Fisher Scientific, Waltham, MA Cat# A-21449 RRID:AB_2535866), anti-Mouse IgG2b-DyLight405 (1:100, Jackson ImmunoResearch Labs Cat# 115-475-207 RRID:AB_2338801), anti-Rabbit IgG-Alexa647 (1:100, Jackson ImmunoResearch Labs Cat# 711-606-152 RRID:AB_2340625); *FACS labeling*: endogenous GFP signal from Atoh1-GFP KI mice, anti-CD15-Dylight649 (1:50, Novus, Novus Biologicals, Littleton, CO Cat# NB100-2672C RRID:AB_1724222); *Co-IP*: anti-GFP (2 μg per IP, [N86/8], UC Davis/NIH NeuroMab Facility Cat# 75–131 RRID:AB_10671445), anti-Jak2 (5 μg per IP, Cell Signaling Technology Cat# 3230 RRID:AB_2128522).

## Cell culture

The following cell lines were purchased from ATCC (Manassas, VA): DAOY (ATCC Cat# HTB-186, RRID:CVCL_1167), 293T (ATCC Cat# CRL-3216, RRID:CVCL_0063). UW228 (RRID:CVCL_4460) (*Huang et al., 2005*) and ONS-76 (Japanese Collection of Research Bioresources Cell Bank, Japan Cat# IFO50355, RRID:CVCL_1624) (*Sun et al., 2013*) were kindly provided by Dr. Charles G. Eberhart (John Hopkins University, Baltimora, MD). All cell lines were verified using visual inspection and

comparison to previously published studies. Cells were found to be negative for mycoplasma contamination. Lines were cultured as adherent cells in DMEM containing 10% FBS and antibiotics using standard cell culture practices (*Geraghty et al., 2014*). For proliferation analysis cells were cultured as medullospheres (*Zanini et al., 2013*) for 3 days prior to Atoh1 induction by addition of doxycycline [CAS: 24390-14-5, 0.2 µM] for 24 hr. Short-term culture of Shh-type MB cells was identical to cerebellar granule cell culture as described previously (*Gao et al., 1991*). Inhibition of Jak2 canonical downstream cascades was as described using SH-4–54 [CAS: 1456632-40-8, 0.5 µM, STAT (*Haftchenary et al., 2013*)], LY294002 [CAS: 154447-36-6, 50 µM, PI3K (*Maira et al., 2009*)], PD184352 [CAS: 212631-79-3, 1 µM, MAPK (*Sebolt-Leopold et al., 1999*)], or AG490 [CAS: 133550-30-8, 5 µM, Jak2 (*Kobayashi et al., 2015*)]. All chemicals were purchased from Selleckchem, Houston, TX.

## Immunoprecipitation-Mass spectrometry (IP-MS)

A total of $10^8$ cells from Atoh1-positive/CD15-positive and Atoh1-positive/CD15-negative cell populations of Nd2::SmoA1$^{Tg}$, Atoh1$^{Atoh1-GFP/Atoh1-GFP}$ tumors were used for immunoprecipitation. Cells were collected and lysed in lysis buffer (50 mM Tris, pH 8.1, 137.5 mM NaCl, 2 mM EDTA, 0.5% Triton-X-100, supplemented with fresh protease inhibitors and phosphatase inhibitors), homogenized using 1 ml insulin syringes. GFP-trap beads (20 µl per IP, ChromoTek, Germany) were added to cleared supernatant and immunoprecipitation reactions were performed overnight at 4°C with gentle head-to-tail rotation. After three washes, immune-complexes were eluted with glycine, vacuum dried and dissolved in 50 mM ammonium bicarbonate, pH = 10 for MS. MS was performed by Proteomics and Metabolomics Core Labs at BCM using standard protocols (*Jung et al., 2017*). We considered a true positive interactor as being able to be pulled down at least in two independent experiments with at least five unique peptides.

## In vitro proliferation assay

Cell proliferation assay was performed on medullospheres of engineered cell lines using CellTiter 96 Non-Radioactive Cell Proliferation Assay (Promega, Madison, WI) as per manufacturer's protocols.

## In vivo tumor growth assay

Xenografts were generated in immunocompromised mice (NOD/SCID) previously described (*Morton and Houghton, 2007*). Cells used were (i) Nd2::SmoA1$^{Tg}$, Atoh1$^{Atoh1-GFP/Atoh1-GFP}$ tumors FACS sorted for Atoh1-positive/CD15-positive, Atoh1-positive/CD15 negative and Atoh1-negative cells (*Figure 1—figure supplement 1b*); (ii) the human medulloblastoma cell lines DAOY (Atoh1-GFP$^{TG}$, Atoh1-GFP Y78F$^{TG}$, Atoh1-GFP Y78D$^{TG}$, *Figure 2c*); or (iii) Nd2::SmoA1$^{Tg}$, Atoh1$^{Atoh1-GFP/Atoh1-GFP}$ tumors infected with shCON, shJak2#58, shJak2#61 viral particles (*Figure 4*). Briefly, $10^5$ cells were mixed in 100 µl cold matrigel (BD Bioscience, San Jose, CA) and injected subcutaneously into the flank region of nude mice (n = 6 per group). Xenografts were left to grow for up to 1 months then harvested and weighed. When the inducible cell lines were used mice were fed doxycycline chow (200 mg/kg, Bio-Serv,) ad libitum throughout the experiment.

## Mathematics and statistical analyses

Statistics were performed using Prism 7.0 (GraphPad Software, La Jolla, CA). Half-life was calculated based on normalized densitometric values obtained by ImageJ and solved for x when y = 0.5 for the following formula: $y = a(e^{-bx})$ as previously described (*Rousseaux et al., 2016*). Animal feedings, treatments and immunofluorescence analyses were performed in a single-blinded fashion. No blinding was used for the remaining analyses. The group size was determined based on previous studies in our laboratories. No animal was excluded. No data points were excluded from the statistical analyses, and variance was similar between the groups being statistically compared. For complete statistical analyses, please refer to *Figure 1—source data 1*. All data presented are of mean ± standard error of the mean (s.e.m.) *, **, *** and **** denote $p < 0.05$, $p < 0.01$, $p < 0.001$, and $p < 0.0001$, respectively. ns denotes $p > 0.05$.

## Data availability

The data sets supporting the conclusions of this article are included within the article and its additional files. Plasmids are available upon reasonable request.

## Acknowledgements

Financial support for this work was provided by the Cancer Prevention and Research Institute of Texas (CPRIT, RP110390) (TJK), the Christopher Getch Fellowship from the Congress of Neurological Surgeons (AJP), the Mary Alice Smith Charitable Foundation (HYZ), and the Howard Hughes Medical Institute (HYZ). The project was supported in part by IDDRC grant number 1U54 HD083092 from the Eunice Kennedy Shriver National Institute of Child Health and Human Development; The Neurovisualization Core (D3 – Microscopy Core). We would like to thank Dr. Charles G Eberhart for providing UW228 and ONS-76 cell lines; Blake Simmons and Rusla du Breuil from transOMIC technologies for their collaboration on RFP-tagged shRNA vectors; Vicky Brandt and members of the Klisch, Zoghbi and Patel laboratories for valuable insight and comments on the manuscript; Stephan Hardin and Yi Huang for excellent mouse husbandry work.

## Additional information

### Competing interests

Huda Y Zoghbi: Senior editor, *eLife*. The other authors declare that no competing interests exist.

### Funding

| Funder | Grant reference number | Author |
| --- | --- | --- |
| Cancer Prevention and Research Institute of Texas | RP110390 | Tiemo J Klisch |
| Congress of Neurological Surgeons | Christopher Getch Fellowship | Akash J Patel |
| Howard Hughes Medical Institute | | Huda Y Zoghbi |
| Mary Alice Smith Charitable Foundation | | Huda Y Zoghbi |

The funders had no role in study design, data collection and interpretation, or the decision to submit the work for publication.

### Author contributions

Tiemo J Klisch, Conceptualization, Formal analysis, Supervision, Funding acquisition, Investigation, Writing—original draft, Writing—review and editing; Anna Vainshtein, Investigation, Writing—review and editing; Akash J Patel, Investigation; Huda Y Zoghbi, Conceptualization, Funding acquisition, Writing—review and editing

### Author ORCIDs

Tiemo J Klisch (iD) http://orcid.org/0000-0001-8182-384X
Anna Vainshtein (iD) http://orcid.org/0000-0003-0368-5044
Akash J Patel (iD) http://orcid.org/0000-0002-1280-2989
Huda Y Zoghbi (iD) http://orcid.org/0000-0002-0700-3349

### Ethics

Human subjects: All patients provided written informed consent and tissues were collected under an IRB approved protocol at Baylor College of Medicine (BCM). Protocol number H-35355.
Animal experimentation: All procedures were approved in advance under the guidelines of the Center for Comparative Medicine, Baylor College of Medicine and were performed in accordance with

the National Institutes of Health Guide for the Care and Use of Laboratory Animals. Protocol number AN-5693.

### Decision letter and Author response

Decision letter https://doi.org/10.7554/eLife.31181.020
Author response https://doi.org/10.7554/eLife.31181.021

## Additional files

### Supplementary files

• Transparent reporting form
DOI: https://doi.org/10.7554/eLife.31181.016

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
