## [Decision Letter]

Thank you for submitting your article "Jak2-mediated phosphorylation of Atoh1 is critical for medulloblastoma growth" for consideration by *eLife*. Your article has been reviewed by three peer reviewers, one of whom is a member of our Board of Reviewing Editors and the evaluation has been overseen by Charles Sawyers as the Senior Editor. The following individuals involved in review of your submission have agreed to reveal their identity: Scott Pomeroy (Reviewer #2).

The reviewers have discussed the reviews with one another and the Reviewing Editor has drafted this decision to help you prepare a revised submission.

Summary:

The report describes the role of the transcription factor Atoh1 (crucial for the normal development of cerebellar granule neurons) in the sonic hedgehog (shh)-driven subtype of medulloblastoma (MB). Using a genetic cross in mice, they initially demonstrated that heterozygous disruption of Atoh1 reduces the tumor burden resulting from shh-driven MB. They then determined that tumor-initiating cells within MB tumors demonstrate tyrosine phosphorylation on Y78 of Atoh1. They also showed that this phosphorylation is mediated by the JAK2 kinase and leads to stabilization of Atoh1 protein and increased transcriptional activity in a reporter assay. As expected, based on these observations, JAK2-specific shRNAs and a pharmacologic JAK inhibitor were found to reduce the tumorigenicity of grafted MB cells in mice. Demonstration that Y78 Atoh1 could be found in human MB tumors further supports the potential clinical relevance of the findings. Overall, this is a well-executed and well-documented study that implicates phosphorylation of Atoh1 on Y78 in the shh-driven subtype of medulloblastoma. As described below, the conclusion that JAK2 is the specific mediator of this phosphorylation event is less definitive.

Essential Revisions:

1) A limitation of the analysis is the lack of Jak2 deletion studies to show whether Jak2 is truly required for medulloblastoma growth. AG490, the inhibitor used to test the role of Jak2 is not highly specific. Additional targets affected by AG490 include EGFR, guanylyl and adenyl cyclases, and Hif1alpha (PMID 22709000, 15209521, 1676428). This lack of specificity complicates the conclusion that Jak2 plays a critical role in medulloblastoma progression. Ideally, the authors could breed a medulloblastoma-causing mutation into a Jak2 conditional background. Conditionally deleting Jak2 in the medulloblastoma cells of origin would test whether other kinases can assume the function of Jak2 when Jak2 is not present. For example, even if Jak1 does not phosphorylate Atoh1 in the presence of Jak2, Jak1 may nevertheless be able the phosphorylate Atoh1 if Jak2 is deleted. The best way to detect this type of complementation is to knock out Jak2 in medulloblastoma-prone mice and then see tumors grow and if Atoh1 gets phosphorylated. Without such genetic data, a firm conclusion should not be drawn that Jak2 is a critical node in medulloblastoma growth. However, if analyzing Jak2 conditional knock out mice is not feasible, the authors should consider focusing instead on the finding that Atoh1 phosphorylation seems to be very important for ATOH1 protein stability, and for medulloblastoma growth. The focus could then be on targeting Atoh1 phosphorylation, which the data strongly suggest would be an effective approach to treatment. Since AG490 decreases phosphorylation of Atoh1, a study testing for decreased tumor growth and increased survival in ND2:SmoA1 mice treated with AG490 could potentially identify a new direction for therapy, even if Jak2 is not necessarily the only target disrupted by the inhibitor.

2) The characterization of the consequences of phosphorylation on Atoh1 activity is a little confusing. The authors claim that the effect of phosphorylation is to stabilize the protein; however, they go on to show in transfected cells using the various phospho-inactive, phospho-mimicking mutants that even though they are expressed at similar levels, the phosphorylation drives increased transcriptional activity in the reporter assay. So, the reader is left to wonder whether phosphorylation is specifically affecting protein levels to drive increased expression or if phosphorylation is more directly affecting Atoh1's activity as a transcription factor. This should be clarified.

3) Related to the previous point, the authors should determine whether phosphorylation affects nuclear localization of Atoh1, which is a level of regulation of transcription factors that is frequently associated with phosphorylation.

4) It would be useful to see that the JAK inhibitor affects Atoh1 half-life and/or transcriptional activity. The authors have relied largely on ectopic expression of Atoh1 mutants to support this connection, but inhibition of endogenous JAK2 to affect endogenous Atoh1 would provide a more compelling link.

5) Despite the complexities that come from understanding Atoh1 within the context of multiple developmental molecular mechanisms, the authors provide minimal data to show Atoh1 expression and activation within the SHH subtype, essentially limited to 2 SHH tumors and 2 Wnt tumors. The manuscript would be strengthened by showing expression in a larger number of tumors, including tumors from Groups 3 and 4 medulloblastoma. There are a number of transcriptome databases that could easily be queried for mRNA expression. A larger number of samples should be assayed for protein expression.

6) The cell lines used in this project have been shown to not be representative of primary medulloblastomas, especially of the SHH subtype. There are a number of groups that have established early passage cultures of primary tumors of all subtypes that can be considered for these experiments. Moreover, for the cell lines used data should be shown of endogenous Atoh1 expression or Y78 phosphorylation.

---

## [Author Response]

Summary:The report describes the role of the transcription factor Atoh1 (crucial for the normal development of cerebellar granule neurons) in the sonic hedgehog (shh)-driven subtype of medulloblastoma (MB). Using a genetic cross in mice, they initially demonstrated that heterozygous disruption of Atoh1 reduces the tumor burden resulting from shh-driven MB. They then determined that tumor-initiating cells within MB tumors demonstrate tyrosine phosphorylation on Y78 of Atoh1. They also showed that this phosphorylation is mediated by the JAK2 kinase and leads to stabilization of Atoh1 protein and increased transcriptional activity in a reporter assay. As expected, based on these observations, JAK2-specific shRNAs and a pharmacologic JAK inhibitor were found to reduce the tumorigenicity of grafted MB cells in mice. Demonstration that Y78 Atoh1 could be found in human MB tumors further supports the potential clinical relevance of the findings. Overall, this is a well-executed and well-documented study that implicates phosphorylation of Atoh1 on Y78 in the shh-driven subtype of medulloblastoma. As described below, the conclusion that JAK2 is the specific mediator of this phosphorylation event is less definitive.Essential Revisions:1) A limitation of the analysis is the lack of Jak2 deletion studies to show whether Jak2 is truly required for medulloblastoma growth. AG490, the inhibitor used to test the role of Jak2 is not highly specific. Additional targets affected by AG490 include EGFR, guanylyl and adenyl cyclases, and Hif1alpha (PMID 22709000, 15209521, 1676428). This lack of specificity complicates the conclusion that Jak2 plays a critical role in medulloblastoma progression. Ideally, the authors could breed a medulloblastoma-causing mutation into a Jak2 conditional background. Conditionally deleting Jak2 in the medulloblastoma cells of origin would test whether other kinases can assume the function of Jak2 when Jak2 is not present. For example, even if Jak1 does not phosphorylate Atoh1 in the presence of Jak2, Jak1 may nevertheless be able the phosphorylate Atoh1 if Jak2 is deleted. The best way to detect this type of complementation is to knock out Jak2 in medulloblastoma-prone mice and then see tumors grow and if Atoh1 gets phosphorylated. Without such genetic data, a firm conclusion should not be drawn that Jak2 is a critical node in medulloblastoma growth. However, if analyzing Jak2 conditional knock out mice is not feasible, the authors should consider focusing instead on the finding that Atoh1 phosphorylation seems to be very important for ATOH1 protein stability, and for medulloblastoma growth. The focus could then be on targeting Atoh1 phosphorylation, which the data strongly suggest would be an effective approach to treatment. Since AG490 decreases phosphorylation of Atoh1, a study testing for decreased tumor growth and increased survival in ND2:SmoA1 mice treated with AG490 could potentially identify a new direction for therapy, even if Jak2 is not necessarily the only target disrupted by the inhibitor.

While using a Jak2 conditional knock out mouse model to delete Jak2 after the Shh-derived tumor has formed is an ideal experiment, this will take several months. To insure the specificity of Jak2 effects, we opted to use viral knock down of Jak2 in primary Shh-derived medulloblastoma tumors. Using xenografts, we show that downregulation of Jak2 indeed impedes tumor growth (Figure 4). Moreover, by western blot analysis we show that Y78 phosphorylation of Atoh1 is absent in the tumors with the most effective Jak2 knock down. Together with the chemical inhibition of Jak2 in cell culture, the notion that Y78 phosphorylation itself increases tumor growth (Figure 2) and the epistatic experiments in which Jak2 alone is sufficient to increase Atoh1 half-life (Figure 3—figure supplement 2) and transcriptional activity (Figure 3—figure supplement 3), we hope that the reviewers will agree that the data do support the conclusion that Jak2 increases tumor growth in vivo through Y78 phosphorylation of Atoh1.

A second point raised pertains to whether another Jak family member, namely Jak1, could take over the role of Jak2 in its absence. While the in vitro kinase assay (Figure 3—figure supplement 1) and the BiFC assay (Figure 3—figure supplement 1) do not suggest such a possibility, the situation may differ in vivo. We addressed this question in two ways. If Jak1 compensates for the loss of function of Jak2, we predict to detect a difference in Atoh1 transcriptional activity in cell lines after Jak2 knock down. Because the chemical inhibitor might not be selective enough and could also inhibit Jak1, we performed a combined genetic and chemical inhibition experiment to address this question. We inhibited Jak2 using AG490 while virally knocking down the protein at the same time, to see if we get an additive effect. To this end, we repeated the luciferase assay in DOAY cells that co-expressed Atoh1 and Jak2 (which resulted in a similar induction as reported in Figure 3) and inhibited Jak2 either via AG490 treatment, via introduction of Jak2 shRNAs (#61 in Figure 4, which has the highest knock down efficiency), or by using a combination of AG490 inhibition and Jak2 shRNA knock-down. The figure below shows that while either inhibition of Jak2 (chemical or genetic) results in a substantial reduction in Atoh1-induced transcriptional activity, the combination does not have an additional effect. These data support the conclusion that the effect of AG490 treatment is mediated by Jak2 inhibition and that Jak1 is not compensating for the lack of Jak2 function when it comes to phosphorylating Y78 on Atoh1 which, in turn, mediates its enhanced transcriptional activity. We are happy to add this experiment to the paper as a supplemental piece of data if the reviewer and editor believe it would be helpful.

**Author response image 1. respfig1:** Jak2 inhibition either by AG490 or knock down using shRNA (#61) similarly prevents the increase in transcriptional activity of an Atoh1-specific reporter in DAOY cells. Combining the inhibitor with genetic knock down does not result in any further inhibition of Atoh1 activity, consistent with Jak2 being the major component responsible for the modulation of Atoh1 (n=36 in duplicates, One-way ANOVA compared to Y78, Dunnett's multiple comparisons test. ****p<0.0001, ns=not significant).

In addition, we analyzed the Jak1 levels in our in vivo knock down of Jak2 (Figure 4). While we detect Jak1 in the control tumors as well as the Jak2 knock down tumors, Jak1 levels do not significantly increase. In addition – although Jak1 is present in these tumors the Y78 phosphorylation is nonetheless dramatically decreased. We included these findings in the Western blot analysis of Figure 4 and modified the text accordingly.

2) The characterization of the consequences of phosphorylation on Atoh1 activity is a little confusing. The authors claim that the effect of phosphorylation is to stabilize the protein; however, they go on to show in transfected cells using the various phospho-inactive, phospho-mimicking mutants that even though they are expressed at similar levels, the phosphorylation drives increased transcriptional activity in the reporter assay. So, the reader is left to wonder whether phosphorylation is specifically affecting protein levels to drive increased expression or if phosphorylation is more directly affecting Atoh1's activity as a transcription factor. This should be clarified.

We show that the phosphorylation of tyrosine 78 on Atoh1 has two effects: an aberrant increase in Atoh1 transcriptional activity as well as a stabilizing effect on the Atoh1 protein. We do not believe that these two effects are in a linear cascade where higher stability leads to higher transcriptional activity as the steady state levels in the cell lines are comparable. It will be interesting to investigate whether phosphorylation effects the On- or Off-rates of Atoh1 on DNA, thereby leading to a longer promoter occupancy, however these experiments are beyond the scope of this paper. We edited the text to make it clear that these two effects are not connected in a linear manner.

3) Related to the previous point, the authors should determine whether phosphorylation affects nuclear localization of Atoh1, which is a level of regulation of transcription factors that is frequently associated with phosphorylation.

This is an excellent suggestion and we agree that this information strengthens the paper. In response, we evaluated the nuclear localization of the wildtype, phospho-mutant and phospho-mimetic Atoh1 constructs and saw no differences. We included the results in Figure 2—figure supplement 1.

4) It would be useful to see that the JAK inhibitor affects Atoh1 half-life and/or transcriptional activity. The authors have relied largely on ectopic expression of Atoh1 mutants to support this connection, but inhibition of endogenous JAK2 to affect endogenous Atoh1 would provide a more compelling link.

Because Atoh1 is not only a proliferation factor but also induces numerous differentiation cascades, there is no cell line (to our knowledge) that expresses endogenous Atoh1. To identify a neuronal cell line that expresses endogenous Atoh1 we interrogated the Swedish Human Protein Atlas with 56 cell lines catalogued (HPA, https://www.proteinatlas.org/cell) as well as the Broad Institute Cancer Cell Line Encyclopedia with 1457 cell lines catalogued (CCLE, https://portals.broadinstitute.org/ccle), but sadly failed to identify any. This is the primary reason for using the inducible expression system, which gives us the advantage of studying the effects of different mutants without the interference of the endogenous protein.

While not ideal, we wanted to address the reviewer’s comment to evaluate the effects on *endogenous* Atoh1 by using mouse primary Shh-derived medulloblastoma cells, which express endogenous Atoh1. We have already used a short culture experiment to show a dramatic reduction in Y78 levels within one hour of AG490 inhibition (Figure 3). We repeated this experiment (n=3) with and without the Jak2 inhibitor while blocking translation (using addition of cycloheximide) at the same time. As shown below, while Atoh1 is stable over at least 8 hours after plating, addition of the Jak2 inhibitor rapidly leads to the loss of Jak2 as well as tyrosine 78 phosphorylation. At the same time, Atoh1 protein stability decreases only in the AG490-treated cells but not in those that were untreated. These data support the conclusion that our cell line data reflect the endogenous state of Atoh1 in SHH-type medulloblastoma cells. We are happy to add this experiment to the paper as a supplemental piece of data if the reviewer and editor believe it would be helpful.

**Author response image 2. respfig2:** Inhibition of Jak2 by AG490 results in the destabilization of Atoh1 in primary Shh-type medulloblastoma cells. Isolated primary mouse Shh-type medulloblastoma cell were plated in a 24-well format and treated with Cyclohexamide (CHX) three hours after plating for the indicated durations. Another set of cells were additionally treated with AG490 to inhibit Jak2 activation. AG490 treatment resulted in a dramatic reduction in Jak2 and tyrosine 78 phosphorylation within 2 hrs, as well as a reduction in Atoh1 stability. n=3 in duplicates.

5) Despite the complexities that come from understanding Atoh1 within the context of multiple developmental molecular mechanisms, the authors provide minimal data to show Atoh1 expression and activation within the SHH subtype, essentially limited to 2 SHH tumors and 2 Wnt tumors. The manuscript would be strengthened by showing expression in a larger number of tumors, including tumors from Groups 3 and 4 medulloblastoma. There are a number of transcriptome databases that could easily be queried for mRNA expression. A larger number of samples should be assayed for protein expression.

Atoh1 expression is almost undetectable in other medulloblastoma subtypes other than the Shh subtype. Accordingly, we re-reviewed existing expression studies and found Atoh1 to be expressed in SHH-type tumors but no other subtypes. To illustrate the exclusivity of Atoh1 expression in SHH-type medulloblastoma, we included a box plot of Atoh1 expression using the publicly available data from Dr. Pfister and colleagues (Kool et al., 2012). While our aim is to test Atoh1 protein levels in human medulloblastoma, we have yet to develop or identify an antibody that reliably detects Atoh1. We and others have tried for years to develop a reliable and specific antibody to no avail, for this reason we created the Atoh1-GFP fusion knock-in allele in the mice. While we acknowledge the limitations of our human data due to the mentioned antibody and sample availability, we believe that our in vivo mouse work is solid in supporting a Jak2-Atoh1 signaling axis in Shh-type medulloblastoma and our human data sufficiently support the notion that Atoh1 is phosphorylated on Y78 in human SHH-type medulloblastoma.

6) The cell lines used in this project have been shown to not be representative of primary medulloblastomas, especially of the SHH subtype. There are a number of groups that have established early passage cultures of primary tumors of all subtypes that can be considered for these experiments. Moreover, for the cell lines used data should be shown of endogenous Atoh1 expression or Y78 phosphorylation.

We take an active part in the brain tumor program at Texas Children’s in Houston. While Dr. Xiao-Nan Li and colleagues established patient tumor-derived orthotopic xenograft models for childhood brain tumors, there is a consensus that newly isolated medulloblastoma cell lines do not grow in culture (Neumann et al., 2017).

Moreover, while there is considerable discussion in the field about the origins and subtype of the different medulloblastoma cell lines used in this study, we would like to highlight for the reviewer a study by Dr. Fanayan and colleagues from earlier this year. This study provides a convincing argument that DAOY cells do have a SHH-type proteomic signature, whereas the UW228 cells display a signature more similar to the WNT subtype (Higdon et al., 2017). This classification differs somewhat from earlier attempts to classify existing cell lines (reviewed in Neumann et al., 2017). It is noteworthy that DAOY cells originated from a desmoplastic cerebellar medulloblastoma, which is a hallmark of SHH-type medulloblastoma.

As mentioned earlier, Atoh1 is not expressed in any of these cell lines probably due to its role in cellular differentiation (Klisch et al., 2011). Therefore, our doxycycline-inducible system provides an opportunity to better study Atoh1 function. While we agree that these medulloblastoma cell lines do not fully recapitulate the tumor biology of medulloblastoma, it is noteworthy that this is an inherent flaw of all in vitro systems. Acknowledging this limitation, we opted to use three rather than just one cell line as well as in vivo studies to provide further support for our conclusions.